# Keel Bone Damage in Laying Hens—Its Relation to Bone Mineral Density, Body Growth Rate and Laying Performance

**DOI:** 10.3390/ani11061546

**Published:** 2021-05-25

**Authors:** Christin Habig, Martina Henning, Ulrich Baulain, Simon Jansen, Armin Manfred Scholz, Steffen Weigend

**Affiliations:** 1Institute of Farm Animal Genetics, Friedrich-Loeffler-Institut, 31535 Neustadt, Germany; mdh55@t-online.de (M.H.); ulrich.baulain@fli.de (U.B.); simon.jansen@fli.de (S.J.); steffen.weigend@fli.de (S.W.); 2Livestock Center of the Faculty of Veterinary Medicine, Ludwig-Maximilians-University Munich, 85764 Oberschleissheim, Germany; armin.scholz@lvg.vetmed.uni-muenchen.de

**Keywords:** keel bone, body weight, housing system, egg production, phylogeny, Gallus gallus, dual energy X-ray absorptiometry

## Abstract

**Simple Summary:**

Keel bone damage is an important welfare issue for laying hens. Four lines of laying hens, differing in phylogenetic origin and laying rate kept in single cages or a floor housing system were weighed and deformities of the keel bone were evaluated regularly between 15 and 69 weeks of age. Deformities, fractures and the bone mineral density of the keels were assessed after hens were euthanized. We analyzed the relationship between bone mineral density and total egg number as well as body growth. Hens kept in cages showed more deformities, but fewer fractures and a lower bone mineral density of the keel bone than did floor-housed hens. Keel bones of white-egg layers had a lower mineral density and were more often deformed compared with brown-egg layers. Keel bones were more often broken in hens of the layer lines with a high laying rate compared to the lines with a moderate laying rate. Laying rate and adult body weight had an effect on the keel bone mineral density. The study contributes to the understanding of factors causing keel bone damage in laying hens. It showed that the bone mineral density greatly affects keel bone deformities.

**Abstract:**

Keel bone damage is an important animal welfare problem in laying hens. Two generations of four layer lines, differing in phylogenetic background and performance level and kept in single cages or floor pens were weighed and scored for keel bone deformities (KBD) during the laying period. KBD, keel bone fractures (KBF) and the bone mineral density (BMD) of the keels were assessed *post mortem*. For BMD, relationships to laying performance and body growth were estimated. Caged hens showed more deformities, but fewer fractures and a lower BMD of the keel bone than floor-housed hens. White-egg layers had a lower BMD (0.140–0.165 g/cm^2^) and more KBD than brown-egg layers (0.179–0.184 g/cm^2^). KBF occurred more often in the high-performing lines than the moderate-performing ones. However, in the high-performing lines, BMD was positively related to total egg number from 18 to 29 weeks of age. The adult body weight derived from fitted growth curves (Gompertz function) had a significant effect (*p* < 0.001) on keels’ BMD. The study contributes to the understanding of predisposing factors for keel bone damage in laying hens. It showed that the growth rate has a rather subordinate effect on keels’ BMD, while the BMD itself greatly affects KBD.

## 1. Introduction

Keel bone damage is an important animal welfare issue, as an alarmingly high number of laying hens show keel bone deformities (KBD) and/or -fractures (KBF) (reviewed by [1]). Moreover, keel bone damages are of economic importance. KBF are often accompanied by a lower egg production [2,3] and poor egg quality including reduced shell thickness and breaking strength [4] as well as lower shell weights [2]. Chargo et al. [5] attributed the decreased production to changes in the hens’ behavior due to pain or musculature changes as a result of keel bone damage. KBF are likely to induce chronic stress and a depression-like state in laying hens [6] and also increase flock mortality [7]. Keel bone damage is a complex trait and there are numerous factors, including nutritional, environmental and genetic ones, which influence the risk and possibility of developing such disorders [8]. KBF often occur as a result of falls within alternative housing systems [9]. In this context, collisions with housing equipment, especially perches, or due to social interactions between hens were identified as a main source of keel bone damage in laying hens [9]. Nevertheless, KBF can also be a result of non-collision events such as wing-flapping [8]. In contrast to these short-duration traumata responsible for KBF, KBD are a result of a protracted process of bone remodeling due to a continued mechanical pressure load during perching [10,11] and therefore increase when giving hens access to perches [12,13].

As selection of laying hens mainly focused on a high number of saleable eggs for many years, skeletal problems, including keel bone disorders, are often considered a direct consequence of the decades-long breeding work. A gradual but persistent loss of structural bone tissue in favor of meeting the calcium requirement for eggshell formation is thought to be the main cause of the increased KBF frequency in layers [14]. However, the assumption that selection for high egg numbers is solely responsible for bone weakness is viewed critically and recent studies suggested, for example, that early puberty is a decisive factor [15,16]. Leyendecker et al. [17] concluded that the lack of exercise laying hens experience in furnished cages has a higher impact on the development of weak bones than calcium depletion from eggshell formation. Rennie et al. [18] calculated a low negative correlation (−0.16) between egg production and trabecular bone volume in hens at the end of the laying period, leading to the assumption that the number of eggs has little or no effect on bone quality.

The current study compared the keel bone status of four lines of laying hens differing in two dimensions, laying performance and phylogenetic origin. Furthermore, we aimed to examine whether the bone mineral density of the keel bone is affected by hens’ individual egg production and growth rate.

## 2. Materials and Methods

### 2.1. Chicken Lines

An animal model consisting of four purebred layer lines differing in laying intensity and phylogenetic background was used. While the two white egg layers, WLA and R11, were descended from White Leghorn, the brown egg strains, BLA and L68, originated from Rhode Island Red and New Hampshire, respectively [19]. Within each of the phylogenetic groups, a high performing line (WLA, BLA) was contrasted to a moderate performing ones (R11, L68). WLA and BLA originated from breeding program of Lohmann Breeders GmbH (Cuxhaven, Germany). They have been maintained in a sire rotation program since 2012 and lay approximately 320 eggs per year. R11 and L68 are part of the resource populations of the Institute of Farm Animal Genetics, Friedrich-Loeffler-Institut (Neustadt, Germany) and achieve a laying performance of about 200 eggs per year [20]. For further information about performance data of caged hens examined in the present study, see Jansen et al. [16].

### 2.2. Housing System, Management and Feeding

The first generation was kept from September 2015 to January 2017 and the second from November 2016 to March 2018. After hatch, sex was determined visually by inverting the cloaca of day-old chicks. Female chicks were individually weighed and wing-tagged for identification. During the 15-week rearing period, hens were kept line-separated in four floor housing pens, located within one room. Each pen had a floor space of 24 m^2^, was littered with wood shavings and was equipped with a feeding line and nipple drinkers. Moreover, pullets had access to straw bales and two wooden ladders per pen that were placed in an inclined position at one of the walls. Each ladder had three crossbars at a height of 35 cm, 80 cm and 130 cm and a width of 110 cm and 150 cm, respectively. The perch space per pullet was approximately 6 cm. For further information on lighting regime, climatic conditions and the composition of feeding stuffs, see Appendix A in Jansen et al. [16]. At the end of the 15th week of age, pullets were transferred to the layer house.

In each of the generations, a total of 288 pullets, 72 per layer line, were distributed equally throughout the single cages of a three-tiered system arranged in two rows separated by a grid. The floor space per cage was 2400 cm^2^, and each cage was equipped with a feeding through, a nipple drinker and a round plastic perch (3 cm in diameter), which was slightly elevated at a height of approximately 6.5 cm and arranged transversely to the feeding through.

In addition, another 384 pullets of the same four lines and sire families in each generation were kept line-separated in a floor housing system, which was located in an adjacent barn room. The floor housing system consisted of two side-by-side rows with eight pens each. Half of the pens had a floor space of 4 m^2^ (2 m × 2 m; small pens) and the other half measured 8 m^2^ (4 m × 2 m; large pens). They were arranged in alternate order and comprised 24 hens each, given a floor space of 1667 cm^2^/hen (small pens) and 3333 cm^2^/hen (large pens), respectively. A droppings pit of 94 cm depth and 67 cm height was located at the longitudinal side of the pens and could be accessed by the hens via jumping or a ramp. One automatic feeder and a line of eight and 13 nipple drinkers was installed on the droppings pit of the small and large pens, respectively, and provided ad libitum food and water supply throughout the whole laying period. Composition of layer diets is given in Appendix A in Jansen et al. [16]. Each pen was equipped with a round metal perch (3.5 cm in diameter) lengthwise over the droppings pit, given approximately 16.5 cm and 8.5 cm perch space pen hen in the large and small pens, respectively. In the large pens, the perch was fixed at a height of 61 cm, and, in the small pens, it was arranged 5 cm above the ground of the dropping pits. To induce further movement, one plastic box was located in the litter area of the large pens. In each pen, hens had access to one nest box of approximately 120 cm length and 80 cm width, which was accessible from the droppings pit via a balcony.

When the hens were moved to the layer house, the lightning period was gradually stepped up from nine hours to 14 h per day in the course of six weeks and remained constant until the end of the laying period.

### 2.3. Data Collection

Chickens of the first generation were weighed every second week from hatch to week 14 on a digital table scale with a weighing accuracy of 0.1 g (Sartorius CPA 16001S, Sartorius Group, Göttingen, Germany). After another weighing immediately before the birds were transferred to the layer house at the end of the 15th week of age, the weighing interval during the laying period was four weeks. In the case of the second generation, the birds were weighed at four-week intervals during the rearing period and at seven time points during the laying period (at the end of week 15, 17, 21, 25, 29, 49, and 69).

Keel bone status of all of the hens of the first generation was assessed in regular intervals, at the end of the 15th, 21st, 29th, 37th, 45th, 57th, 61st, and 69th week of age. According to Habig and Distl [21], the severity of KBD was recorded on a three-scale scoring system (4 = no deformity, 3 = slight deformity, 2 = moderate to severe deformity). The hens were securely held upside down by their legs with one hand, and the keel bone was palpated using the thumb and forefinger of the other hand. Scoring was made by the same experienced person at any time, which was not blinded to the treatment.

In both generations, the laying hens were euthanized by carbon dioxide inhalation after 70 weeks within five consecutive days, with hens randomly selected each day from both housing systems and all genetic lines. The keel bones were dissected, with adherent muscle tissue being removed, and stored frozen at −20 °C for further analysis. The ’small animal’ mode of a GE Lunar *i*DXA scanner (enCORE^®^ software version 13.5, GE Healthcare GmbH, Solingen, Germany) was used to measure bone mineral density (BMD) of the keel bones by dual energy X-ray absorptiometry (DXA). Therefore, the whole keel bones were scanned in a latero-lateral direction. The dissected keel bones were scored by palpation for KBD and KBF. Deformities were evaluated according to the scale mentioned above. Additionally, the direction of the deviation was noted as transverse and/or sagittal. In order to improve detection of fractures, keel bones were placed on a light box. If fractures occurred, the type of fracture (transversal or longitudinal) as well as the number of fractures (1 to 3 or ≥4), and their location in the cranial, intermediate or caudal third of the carina sterna were recorded. Scoring was done by the same experienced examiner, blinded to the housing system.

### 2.4. Statistical Analyses

Statistical analyses were performed using JMP v14.0 (Statistical Analysis System Institute, Cary, NC, USA, 2018) and R v4.0.2 (R Core Team, Vienna, Austria, 2018).

#### 2.4.1. Deformities and Fractures of the Keel Bone

*Intra vitam* KBD scores of the first generation and *post mortem* KBD and KBF scores of both generations, including the direction of KBD and the type, number and location of fractures, were analyzed by Chi-square tests towards an effect of the generation, layer line and housing system. Statistical significance was set at *p* < 0.05.

#### 2.4.2. Approximation of Growth Curves

Growth curves were fitted to the body weight records of the individual hens applying the Gompertz function [22], computing the parameters of asymptotic final weight at the 69th week of age (a), the maximal growth rate (slope of the growth curve (b)) and the time of inflection, when growth rate is starting to decrease (c). We then analyzed the effect of the generation, housing system, layer line and their interactions on the growth parameters according to the following model:(1)$$\gamma_{ijklm}=\mu +G_{i}+HS_{j}+LL_{k}+G_{i}HS_{j}+G_{i}LL_{k}+HS_{j}LL_{k}+S_{l}+\varepsilon_{ijklm}$$
where $\gamma_{ijklm}$ is the respective growth parameter (a, b, or c); μ is the general mean; $G_{i}$ is the fixed effect of the generation (i = 1, 2); $HS_{j}$ is the fixed effect of the housing system (j = 1, 2); $LL_{k}$ is the fixed effect of the layer line (k = 1 to 4); $G_{i}HS_{j}$, $G_{i}LL_{k}$ and $HS_{j}LL_{k}$ are the interactions of the respective variables; $S_{l}$ is the random effect of the sire (l = 1 to 145); and $\varepsilon_{ijklm}$ is the random error variance. Tukey’s HSD (honestly significant difference) test was performed for multiple comparisons of means. Statistical significance was set at *p* < 0.05.

#### 2.4.3. Bone Mineral Densities Measured in Floor Pens and Cages

We applied a linear mixed model to study the variation of the keel BMD considering main factors and the growth curve parameters as covariates. The model was as follows:(2)$$\begin{array}{c} \gamma_{ijklmnopqr}=\mu +G_{i}+HS_{j}+LL_{k}+KBD_{l}+KBF_{m}+G_{i}HS_{j}+G_{i}LL_{k}+HS_{j}LL_{k}+G_{i}KBD_{l}+G_{i}KBF_{m}+\\ LL_{k}KBD_{l}+LL_{k}KBF_{m}+HS_{j}KBD_{l}+HS_{j}KBF_{m}+KBD_{l}KBF_{m}+a_{n}+b_{o}+c_{p}+S_{q}+\varepsilon_{ijklmnopqr}\;\;\;\;\;\;\;\;\;\;\; \end{array}$$
where $\gamma_{ijklmnopqr}$ is the observation of BMD; μ is the general mean; $G_{i}$ is the fixed effect of the generation (i = 1, 2); $HS_{j}$ is the fixed effect of the housing system (j = 1, 2); $LL_{k}$ is the fixed effect of the layer line (k = 1 to 4); $KBD_{l}$ is the effect of *post mortem* keel bone deformities (l = 1 to 3); $KBF_{m}$ is the effect of *post mortem* keel bone fractures (m = 1, 2); $G_{i}HS_{j}$, $G_{i}LL_{k}$, $HS_{j}LL_{k}$, $G_{i}KBD_{l}$, $G_{i}KBF_{m}$, $LL_{k}KBD_{l}$, $LL_{k}KBF_{m}$, $HS_{j}KBD_{l}$, $HS_{j}KBF_{m}$ and $KBD_{l}KBF_{m}$ are the interactions of the respective variables; $a_{n}$ is the effect of the asymptotic final weight of the growth curve (a); $b_{o}$ is the effect of the slope of the growth curve (b); $c_{p}$ is the effect of the inflection point of the growth curve (c); $S_{q}$ is the random effect of the sire (q = 1 to 145); and $\varepsilon_{ijklmnopqr}$ is the random error variance. Tukey’s HSD (honestly significant difference) test was performed for multiple comparisons of means. Statistical significance was set at *p* < 0.05.

#### 2.4.4. Bone Mineral Densities Measured in Single Cages

As an extension of the basic model (2), the individual laying performance was included as additional factor in model (3), which is why it was only applicable to the single cage data. The laying performance was divided into two laying periods (LP), with LP1 covering weeks 18 to 29 (pre-peak) and LP2 covering weeks 30 to 69 (post-peak). We cleaned the dataset by removing hens (i) with total egg numbers outside the line specific threefold interquartile range (IQR) (<X_0.25_ − 3 x IQR; >X_0.75_ + 3 x IQR) and those who did not lay an egg during the last three consecutive weeks of the study (*n* = 52) [16], and (ii) for which no Gompertz growth function could be fitted (*n* = 5). After filtering, a total of 519 individuals (BLA: *n* = 130; L68: *n* = 128; R11: *n* = 133; WLA: *n* = 128) remained for further analyses. The BMD measurements of these hens were analyzed considering the factors given in model (2) as well as the performance data including the age at first egg and the egg number within the two laying periods. The model was as follows:(3)$$\begin{array}{cc} \gamma_{ijklmnopqrst}= & \mu +G_{i}+LL_{j}+KBD_{k}+KBF_{l}+G_{i}LL_{j}+G_{i}KBD_{k}+G_{i}KBF_{l}+LL_{j}KBD_{k}+LL_{j}KBF_{l}+\\ & KBD_{k}KBF_{l}+a_{m}+b_{n}+c_{o}+S_{p}+FE_{q}+LP1_{r}+LP2_{s}+\varepsilon_{ijklmnopqrst} \end{array}$$
where $\gamma_{ijklmnopqrst}$ is the observation of BMD; μ is the general mean; $G_{i}$ is the fixed effect of the generation (i = 1, 2); $LL_{j}$ is the fixed effect of the layer line (j = 1 to 4); $KBD_{k}$ is the effect of *post mortem* keel bone deformities (k = 1 to 3); $KBF_{l}$ is the effect of *post mortem* keel bone fractures (l = 1, 2); $G_{i}LL_{j}$, $G_{i}KBD_{k}$, $G_{i}KBF_{l}$, $LL_{j}KBD_{k}$, $LL_{j}KBF_{l}$ and $KBD_{k}KBF_{l}$ are the interactions of the respective variables; $a_{m}$ is the effect of the asymptotic final weight of the growth curve (a); $b_{n}$ is the effect of the slope of the growth curve (b); $c_{o}$ is the effect of the inflection point of the growth curve (c); $S_{p}$ is the random effect of the sire (p = 1 to 145); $FE_{q}$ is the effect of the age at first egg laid; $LP1_{r}$ is the effect of the number of eggs laid within weeks 18 to 29; $LP2_{s}$ is the effect of the number of eggs laid within weeks 30 to 69; and $\varepsilon_{ijklmnopqrst}$ is the random error variance. Tukey’s HSD (honestly significant difference) test was performed for multiple comparisons of means. Statistical significance was set at *p* < 0.05.

As the pre- (LP1) and post-peak (LP2) egg number as well as the asymptotic final body weight at the 69th week of age had a significant effect on keels BMD, these traits were analyzed within the layer lines, applying a univariate regression approach. The linear relationships between egg number and adult body weight, respectively, and the keel BMD was modelled as follows:(4)$$\gamma_{i}=\beta_{0}+\beta_{1}x_{i}+\varepsilon_{i}$$
where $\gamma_{i}$ is the observation of BMD; $\beta_{0}$ is the intercept; $\beta_{1}$ is the slope; $x_{i}$ is the pre-peak (week 18 to 29) or post-peak (week 30 to 69) egg number or the asymptotic final weight of the growth curve (a); and $\varepsilon_{i}$ is the random error variance.

## 3. Results

### 3.1. Keel Bone Deformities Assessed Intra Vitam in the First Generation

Figure 1 shows the KBD prevalence assessed *intra vitam* in the course of the laying period. The corresponding significant differences in KBD frequencies between layer lines and housing systems within the single weeks of age are given in Table 1. After the rearing period at the end of 15 weeks of age (Figure 1, Part A), none of the L68- and R11-hens showed KBD and only one BLA-pullet had a slight deformity. In contrast, WLA hens showed a significantly higher frequency of KBD of six hens (approx. 3%) affected by a slight to severe deformity. In single cages (Figure 1, Part C), L68 hens were significantly less affected by KBD than BLA hens at any time of examination (Table 1). The same tendency was observed within the floor housing system (Figure 1, Part B), but significant results were only detected for the 69th week of age. While KBD occurred significantly more often and more severe in hens of the white-egg layer lines compared to the brown ones at any time of examination within the cage system, in the floor housing system, the same trend was seen from the 29th week of age onwards, without reaching a significant value at any day of scoring. With the exception of the 21st week of age, where WLA (45.83%) hens kept in cages showed less deformities than R11 (76.39%), no significant differences were detected between the two white-egg layers within the same housing system. In all layer lines, KBD appeared significantly more often with a higher degree of severity in hens housed in cages (Figure 1, Part C) than in floor pens (Figure 1, Part B). The smallest differences between the two housing systems were recorded for L68 and reached a significant level in the 57th and 61st week of age. The compartment size of the floor housing system had no significant effect on the occurrence of KBD.

### 3.2. Post Mortem Examination of Keel Bone Status

Results of *post mortem* keel bone status of laying hens of the first and second generation are presented in Table 2 and Table 3, respectively. As the effect of generation was significant for all of the keel bone traits (data not shown), with the exception of KBF, the *post mortem* keel bone status of the first and second generation were analyzed separately.

In all layer lines, the number of hens with KBD as well as the severity of deformation was significantly higher in cages compared to the floor housing system. Accordingly, the direction of deformities that occurred in floor pens was primarily transverse, while most of the very severe deformities due to a transverse and at the same time sagittal deviation were found in hens kept in cages. Only one R11-hen showed a KBD that was directed sagittal.

In contrast to these findings, hens kept in floor pens were significantly more often affected by KBF (BLA: 61.82%; L68: 28.40%; R11: 39.27%; WLA: 74.83%) than their conspecifics in cages (BLA: 38.79%; L68: 5.15%; R11: 20.71%; WLA: 36.03%), with the exception of R11 in the first generation, where KBF were equally distributed among housing systems. Overall, 38.77% of all scored keel bones (both generations) showed at least one fracture. More than 89% of these fractures were found in the caudal part of the keel bone (carinal apex). In contrast, fractures solely located in the intermediate (4.24%) and cranial (0.42%) part of the keel bone occurred less often. Approximately 6% of the hens with a broken keel bone showed fractures in at least two different locations.

In the first generation, none of the effects were significant regarding the number of fractures at the caudal part of the keel. In contrast, the second generation was different, with WLA hens kept in floor pens having significantly more KBF at the caudal keel portion compared to L68 within the same system and R11 in both housing systems. However, within cages, fractures accounted at the keels’ tip were lower for WLA than for R11, whereas in floor pens the number was higher for BLA compared to R11.

Only one L68- and one R11-hen had a longitudinal fracture in the caudal part of the keel bone, the rest of the fractures detected in this area were transversely directed.

In floor pens as well as in cages, the keel bones of white-egg layers were significantly more often deformed than those of brown-egg lines, with the exception of the comparison of BLA and WLA kept in cages in the second generation. Hens of the BLA line showed significantly more KBD as well as KBF than L68 in both housing systems except for deformities detected in floor pens of the first generation. Similarly, the frequency and/or degree of KBD was significantly higher for the high-performing white-egg strain compared to their moderate-performing counterpart, with the exception of hens kept in cages the second generation. However, within the phylogenetic groups, KBF were more common in the high-performing strains than the moderate-performing ones in both husbandry systems.

The compartment size of the floor housing system had a significant effect on the *post mortem* keel bone status (Appendix A). With the exception of L68 hens kept in the second generation, keel bones of all lines of laying hens were significantly more frequently broken in hens out of the large compartments compared to the small ones. Furthermore, in the two high-performing layer lines BLA (first generation) and WLA (second generation), the number of fractures located in the caudal third of the keel bone was significantly higher and keel bone deformities occurred significantly more often and more severely when hens were kept in the large pens rather than the small pens.

### 3.3. Fitting the Gompertz Function to the Growth Data

The results of fitting the Gompertz function to growth data are shown in the Appendix A. The main factors of generation, layer line, and housing system as well as the interaction between layer line and housing system, had a significant effect on all of the parameters of the Gompertz growth curve. The interaction effect of generation and layer line was significant for the curves’ point of inflection, while the interaction of generation and housing system significantly influenced the slope of the curve.

### 3.4. Bone Mineral Density of the Keel Bone in both Housing Systems

Table 4 shows the effects of the main factors and their interactions on the BMD of the keel bone. The BMD was significantly affected by the fixed effects of generation, layer line, housing system and KBD, the layer line × housing system and housing system × KBD interactions, and the adult body weight at 69 weeks of age. The corresponding least squares means (LSM) for the keels’ BMD are given in Table 5. Across layer lines, the BMD of keel bones examined in hens of the first generation was significantly higher compared to those measured in the second generation. In hens of both generations, a significant effect of the housing system, but not compartment size (data not shown), on BMD of the keel bone was detected. All layer lines showed significantly higher values of keel BMD when they were kept in floor pens than in cages. Across both generations, WLA hens had a significantly lower BMD of the keel bone compared to the other layer lines. Within housing systems, this was also the case in cages, while, within floor pens, R11 and WLA had similar values. While no significant differences have been detected between BLA and L68, the moderate-performing layer line R11 had a significantly higher BMD of the keel bone than their high-performing counterpart WLA when they were kept in cages.

The BMD of the keel bone differed significantly between the three deformity scores. Across all layer lines and housing systems, the BMD of severe deformed keel bones (score 2) was significantly lower compared to slightly (score 3) or undeformed keel bones (score 4). While the BMD did not differ significantly between the three deformity scores when hens were housed in floor pens, all of the scores differed significantly from each other in cages, i.e., the higher the keels’ BMD, the less severe its deformity. In contrast, no significant relationship between KBF and BMD was shown.

Regarding the effects of the Gompertz growth curve parameters, the analysis revealed no significant effect of the slope of the curve and its point of inflection on keels’ BMD, but a significant positive relationship between BMD of the keel bone and the adult body weight in white-egg layers, i.e., heavier birds had a higher BMD of the keel bone.

### 3.5. Bone Mineral Density Measured in Single Cages

Further analyses of the cleaned dataset (519 individuals) of hens kept in cages for relationships between BMD, laying performance, and growth rate showed significant effects of the main factors of generation and KBD as well as the total number of eggs laid during the pre- (LP1) and post-peak (LP2) period and the adult body weight on keels’ BMD (Table 6).

In accordance with the results presented for the keels’ BMD in both housing systems (Table 5), laying hens with severe deformed keel bones showed lower BMD values compared to their conspecifics with slightly or undeformed keel bones (Table 6).

Regarding laying hens’ performance, it could be shown that the age when the first egg was laid had no effect on the BMD of the keel bone. In contrast, the number of eggs laid during the early laying period from 18 to 29 weeks of age was significantly related to the BMD of the keel bone in BLA, whereas the laying performance from 30 to 69 weeks of age, i.e., from the peak until the end of the laying period significantly influenced keel BMD in R11.

Results of univariate regression analysis, which was only performed for those covariates found to significantly influence keels’ BMD (Table 6), are shown in Figure 2. Regression coefficients ($\beta_{1}$) of the total number of eggs laid during the early laying period and from 30 to 69 weeks of age in relation to the BMD of the keel bone are presented in Figure 2A and Figure 2C, respectively. While, in the high-performing layer lines, significant positive regression coefficients were obtained for the total number of eggs laid from 18 to 29 weeks of age (WLA: $\beta_{1}$ = 4.39 × 10^−4^, BLA: $\beta_{1}$ = 0.141 × 10^−^^4^), regression coefficients for the moderate-performing lines were not significant. In contrast, a significant negative regression coefficient ($\beta_{1}$ = 4.16 × 10^−4^) was observed for the laying rate from 30 to 69 weeks of age and keels’ BMD in R11. However, further significant relationships between the BMD of the keel bone and laying rate have not been observed. The trends of BMD with increasing egg number from 18 to 29 weeks of age and from 30 to 69 weeks of age, respectively, are shown in Figure 2B and Figure 2D.

While the slope of the Gompertz growth curve and its point of inflection had no significant effect on the BMD of the keel (Table 6), a significant positive relationship was found for the adult body weight in white-egg layers, i.e., heavier birds had a higher BMD of the keel bone. Figure 2E shows the regression coefficients ($\beta_{1}$) of the adult body weight at 69 weeks of age in relation to the BMD of the keel bone and Figure 2F illustrates the trend of BMD with increasing body weight. A positive, but not significant, relationship was found between the BMD of the keel bone and the adult body weight, with the exception of L68, where the trend was rather negatively directed.

## 4. Discussion

The objective of the present study was to assess keel bone damage in laying hens and its relationship to keel BMD, laying performance, and body growth rate. We detected KBD more often in the white-egg layer lines WLA and R11 compared to the brown-egg layer lines BLA and L68. This is in accordance with the findings of Eusemann et al. [23], who scored nine to ten hens per layer line (BLA, WLA, L68, R11, G11) and housing system (single cages vs. floor housing system) repeatedly in the 35th, 51st, and 72nd week of age for keel bone damage using X-ray images. One possible explanation for the observed differences between white- and brown-egg layers might be that, in white-egg layers, the frequency of perching is higher compared to brown-egg layers [24], in particular when kept in alternative housing systems, rather than in cages. Regarding KBF, the risk-promoting factors causing differences between the two phylogenetic groups are contradictory. On the one hand, white hens’ risk for collisions with the housing equipment and resulting KBF seem to be lower due to better flight and 3D-movement skills than brown hens [25,26]. On the other hand, white-egg layers are more fearful and more flighty than brown-egg layers [27,28], which can cause more panic reactions at the group and individual level [29], and consequently KBF due to collisions with the housing system [8,25]. Although R11-hens kept in the current study were also very nervous and got quite agitated and panicky when persons passed or entered the cages or floor pens (personal observations), their keel bones were significantly less often broken than those of BLA- and WLA-layers. Heerkens et al. [25] assumed that the keel bones of white hens, known to produce more eggs than brown hens, become progressively weaker, raising its vulnerability for deformities. Candelotto et al. [4] examined the likelihood of experimental KBF of four specific cross-bred and one pure line, differing in egg production and quality. Their results implied a strong propensity for genetic regulation of fracture susceptibility, as one of the lines, which has not been selected for egg production for several years had less than 20% of experimental fractures, while more than 90% of the keel bones of hens of another commercial line fractured due to impact testing. Our results are in accordance with the observations made by Candelotto et al. [4,30], as we found less KBD and KBF in the moderate-performing line L68 compared to its high-performing counterpart BLA as well as more severe deformities and a higher fracture prevalence at necropsy in WLA than R11. This is consistent with observations on KBF occurrence from a previous study with hens of the same genetic line [31]. Furthermore, regression analyses revealed a significant negative relationship for keels’ BMD and the number of eggs laid from 30 to 69 weeks of age in R11 hens kept in cages. This agrees with the findings of Jansen et al. [16], who observed decreasing BMDs of the humerus and tibiotarsus with increasing total eggshell production in R11. However, for the pre-peak production, a significant positive relationship was found for the high-performing layer lines, i.e., the higher the number of eggs laid until the 29th week of age, the higher the hens’ keel BMD. In contrast, the moderate-performing lines showed negative regression coefficients for laying performance from 30 to 69 weeks of age, although not significant. These results suggest that, alongside increased egg production, the BMD of laying hens might be also increased following selection and support early findings that a high laying performance does not necessarily adversely affect bone quality [15,16].

The majority of fractures observed in the present study were located in the caudal third, i.e., the tip, of the carina sterna. In accordance with this finding, other researchers [5,28,31] observed high prevalence’s of fractures located at the tip of the keel bone assessed through a three-dimensional model, radiography and CT scans, respectively. This might be due to the fact that this is the last part to be ossified [32,33] and its higher susceptibility to impacts and consequently resulting damages depending on the position of the caudal keel bone portion in the hen’s body [34].

In contrast to the results of the present study, Eusemann et al. [23] did not find a significant difference in the prevalence of KBD and KBF between the housing systems, with the exception of week 72, when the number of hens with fractured keel bones was higher in the floor housing system compared to cages. A possible explanation for this contrary result might be seen in the different methodologies used for the detection of keel bone damages, the lower sample size examined by Eusemann et al. [23] as well as the fact that the housing systems were not totally equal in the two studies. Overall, the housing system is seen as one of the main factors influencing keel bone damage in laying hens. The more complex a housing system is, the higher is the probability of hazardous events resulting in KBF [35]. By reducing falls and collisions, the installation of ramps in aviary systems can limit the damaging consequences for laying hens [36]. As the prevalence of fractures was low compared to several previous studies (reviewed by [37]), it might be assumed that the number of KBF would be higher if floor pens examined in the present study had not been equipped with ramps to give access to the droppings pit. However, a recent study of Thøfner et al. [38] raises doubts about whether collisions are the main cause of KBF. Moreover, it has to be considered that results might be confounded due to the different compartment sizes and perch heights within the floor housing system and the different perch types in the two housing systems examined in the present study.

Apart from the described hazards of a large degree of freedom of movement in alternative husbandry systems, the higher motion activity can lead to an increased bone quality too [17]. Even a higher breaking strength [39] or shear strength of the tibiotarsus can contribute to a reduced number of hens with KBF, likely due to the positive correlation of total bone mineral content of the keel bone and the tibiotarsus [40]. In accordance with previous reports [41], keel bones with a high BMD were less severe deformed when hens were kept in cages. The same tendency was shown for layers housed in floor pens but did not reach significance. Similarly, Candelotto et al. [30] found laying hens being most resistant to experimental KBF having the greatest keel BMD, while the two lines most susceptible to fractures had the lowest values, indicating this property as important in affecting keel bones’ breaking strength. The higher the BMD, the lower the prevalence of keel bone damages, most probably arising from a higher bone breaking strength, as could be shown previously [41,42]. This confirms the results drawn from several other studies. Gebhardt-Henrich et al. [40] detected a greater cortical and trabecular bone mineral content and calcium content in intact or only slightly deformed keel bones than in broken ones. Moreover, they found lower keel BMDs in white-egg compared to brown-egg layers, which follows our findings. Toscano et al. [43] described the BMD of the keels’ lateral surface as an effective predictor of the likelihood of KBF, with an increasing BMD decreased the likelihood of a fracture. Fleming et al. [42] and Stratmann et al. [44] compared keel bone damages in hens of a pure line selected for high bone strength (H line), characterized by a greater BMD, with an line selected for low bone strength (L line) and found fewer KBF and KBD in hens of the H line.

Regarding the effect of the parameters of the Gompertz growth curve on keel BMD, the analysis of variance of data assessed in both housing systems (model 2) and single cages (model 3), respectively, revealed a significant positive relationship between the adult body weight and keels’ BMD, i.e., the heavier the laying hen is, the higher is her keel BMD. However, this significant effect disappeared when within-line univariate regression analyses (model 4) were performed.

## 5. Conclusions

The study contributes to the understanding of predisposing factors for keel bone damage in laying hens. Based on our results, we can confirm that the housing system and age of laying hens affect the occurrence of keel bone alterations. Both performance level and phylogenetic background seem to play a role regarding the development of keel bone damage. The current experiment showed that the adult body weight had an effect on the BMD of the keel bone in laying hens. However, the keels’ BMD was shown to be decisive and greatly affects the susceptibility of the keel bone to be injured.

## Figures and Tables

**Figure 1 animals-11-01546-f001:**
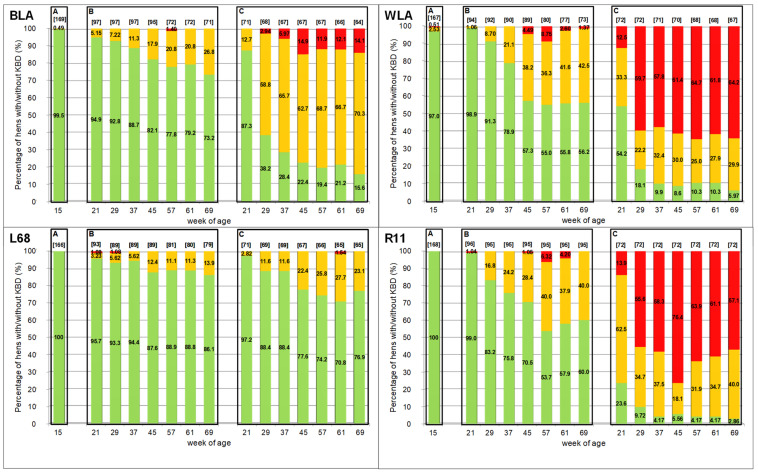
Prevalence of keel bone deformities (KBD) assessed at the end of the rearing period (**A**) and during the laying period, when hens were kept in floor pens (**B**) or single cages (**C**) of the first generation. Each bar represents the percentages of hens without deformities (Score 4; green part of the bar), with slightly deformed (Score 3; yellow part of the bar) and moderate to severe deformed keel bones (Score 2; red part of the bar), respectively. The number of examined hens within the single weeks of age is given in brackets above the bars.

**Figure 2 animals-11-01546-f002:**
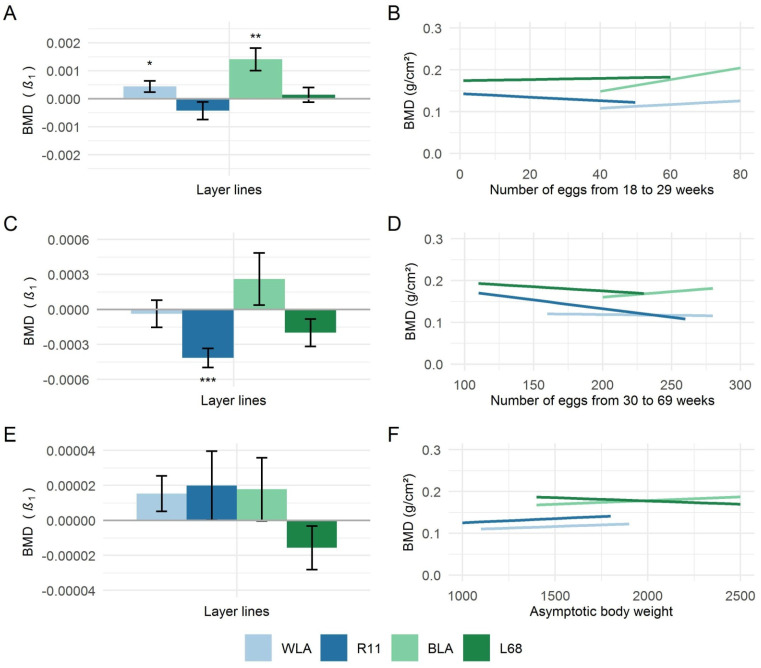
Regression coefficients ($\beta_{1}$) ± standard errors of regression of total number of eggs laid from 18 to 29 weeks of age (**A**), total number of eggs laid from 30 to 69 weeks of age (**C**) and the adult body weight at 69 weeks of age (**E**) pertaining to the bone mineral densities (BMD) of the keel bone, and the effect of total number of eggs laid from 18 to 29 weeks of age (**B**), total number of eggs laid from 30 to 69 weeks of age (**D**), and the adult body weight at 69 weeks of age (**F**) on the BMD of the keel bone in four chicken layer lines (WLA, R11, BLA, L68). Significant regression coefficients are marked with asterisks (* *p* < 0.05, ** *p* < 0.01, *** *p* < 0.001).

**Table 1 animals-11-01546-t001:** Number of hens examined (*n*) and percentage of hens with keel bone deformities (Score 2 and 3) assessed *intra vitam* in the first generation.

Week of Age	Floor Housing								Single Cages							
	BLA		L68		R11		WLA		BLA		L68		R11		WLA	
	*n*	%	*n*	%	*n*	%	*n*	%	*n*	%	*n*	%	*n*	%	*n*	%
15	169	0.49 ^AB^	166	0 ^B^	168	0 ^B^	167	2.99 ^A^	-	-	-	-	-	-	-	-
21	97	5.15 ^D^	93	4.30 ^DE^	96	1.04 ^E^	94	1.06 ^DE^	71	12.68 ^C^	71	2.82 ^DE^	72	76.39 ^A^	72	45.83 ^B^
29	97	7.22 ^D^	89	6.74 ^D^	95	16.84 ^C^	92	8.70 ^CD^	68	61.76 ^B^	69	11.59 ^CD^	72	90.28 ^A^	72	81.94 ^A^
37	97	11.34 ^DE^	89	5.62 ^E^	95	24.21 ^C^	90	21.11 ^CD^	67	71.64 ^B^	69	11.59 ^DE^	72	95.83 ^A^	71	90.14 ^A^
45	95	17.89 ^DE^	89	12.36 ^E^	95	29.47 ^CD^	89	42.70 ^C^	67	77.61 ^B^	67	22.39 ^DE^	72	94.44 ^A^	70	91.43 ^A^
57	72	22.22 ^DE^	81	11.11 ^E^	95	46.32 ^C^	80	45.00 ^C^	67	80.60 ^B^	66	25.76 ^D^	72	95.83 ^A^	68	89.71 ^A^
61	72	20.83 ^EF^	90	11.25 ^E^	95	42.10 ^CD^	77	44.16 ^C^	66	78.79 ^B^	65	29.23 ^C^	72	95.83 ^A^	68	89.71 ^A^
69	71	26.76 ^DE^	79	13.92 ^F^	95	40.00 ^CD^	73	43.84 ^C^	64	84.37 ^B^	65	23.08 ^EF^	70	97.14 ^A^	67	94.03 ^A^

^A–F^ Frequencies within a row with no common superscript differ significantly at *p* < 0.05.

**Table 2 animals-11-01546-t002:** Percentage of dissected keel bones with/without several damages in hens of the first generation.

Trait	Floor Housing				Single Cages			
	BLA	L68	WLA	R11	BLA	L68	WLA	R11
	*n* = 71	*n* = 79	*n* = 73	*n* = 95	*n* = 64	*n* = 65	*n* = 66	*n* = 70
Percentage of keel bones with/without deformities ^1^								
Score 4	70.42	82.28	41.10	47.37	15.63	61.54	3.03	2.86
Score 3	23.94	15.19	38.36	47.37	26.56	33.85	6.06	24.29
Score 2	5.63	2.53	20.55	5.26	57.81	4.62	90.91	72.86
Significance	AB	A	D	C	E	BC	F	G
Direction of keel bone deformity ^2^								
Score 0	0	0	0	1.96	0	0	0	1.49
Score 1	85.71	86.67	65.12	82.35	16.67	76.00	4.69	10.45
Score 2	14.29	13.33	34.88	15.69	83.33	24.00	95.31	88.06
Significance	A	A	A	A	B	A	C	BC
Percentage of keel bones with/without fractures ^3^								
Score 0	33.80	75.64	27.40	71.28	62.50	95.38	53.03	72.86
Score 1	66.20	24.36	72.60	28.72	37.50	4.62	46.97	27.14
Significance	B	E	A	E	DE	F	CD	E
Percentage of numbers of fractures in the caudal third of fractured keel bones								
1	37.78	52.94	43.64	53.85	60.00	66.67	56.00	62.50
2	31.11	11.67	25.45	34.62	25.00	0	16.00	25.00
3	17.78	23.53	12.73	11.54	5.00	33.33	20.00	12.50
≥4	13.33	11.76	18.18	0	10.00	0	8.00	0
Significance	A	A	A	A	A	A	A	A

^1^ Scoring system: 4 = no deformity, 3 = slight deformity, 2 = moderate to severe deformity. ^2^ Scoring system: 0 = sagittal, 1 = transverse, 2 = sagittal and transverse. ^3^ Scoring system: 0 = fracture absent, 1 = fracture present. A–G: Overall frequencies within a row with no common letter differ significantly at *p* < 0.05.

**Table 3 animals-11-01546-t003:** Percentage of dissected keel bones with/without several damages in hens of the second generation.

Trait	Floor Housing				Single Cages			
	BLA	L68	WLA	R11	BLA	L68	WLA	R11
	*n* = 92	*n* = 90	*n* = 74	*n* = 96	*n* = 72	*n* = 71	*n* = 70	*n* = 70
Percentage of keel bones with/without deformities ^1^								
Score 4	68.09	91.11	54.05	70.83	30.56	61.97	11.43	5.71
Score 3	22.34	8.89	28.38	22.92	25.00	30.99	28.57	35.71
Score 2	9.57	0	17.57	6.25	44.44	7.04	60.00	58.57
Significance	BC	A	C	B	D	BC	E	E
Direction of keel bone deformity ^2^								
Score 0	0	0	0	0	0	0	0	1.49
Score 1	38.71	97.50	52.94	60.71	24.00	40.74	19.67	11.94
Score 2	61.29	12.50	47.06	39.29	76.00	59.26	80.33	86.57
Significance	BC	A	AB	AB	CD	BC	D	D
Percentage of keel bones with/without fractures ^3^								
Score 0	41.49	67.78	25.68	50.00	59.72	94.37	74.29	85.51
Score 1	58.51	32.22	74.32	50.00	40.28	5.63	25.71	14.49
Significance	B	DE	A	BC	CDE	G	EF	FG
Percentage of numbers of fractures in the caudal third of fractured keel bones								
1	50.91	71.43	34.55	70.83	58.62	100	43.75	100
2	29.09	25.00	29.09	22.92	24.14	0	31.25	0
3	7.27	0	14.55	6.25	6.90	0	18.75	0
≥4	12.73	3.57	21.82	0	10.34	0	6.25	0
Significance	AB	BCD	A	CD	ABCD	ABCD	AD	BC

^1^ Scoring system: 4 = no deformity, 3 = slight deformity, 2 = moderate to severe deformity. ^2^ Scoring system: 0 = sagittal, 1 = transverse, 2 = sagittal and transverse. ^3^ Scoring system: 0 = fracture absent, 1 = fracture present. A–G: Overall frequencies within a row with no common letter differ significantly at *p* < 0.05.

**Table 4 animals-11-01546-t004:** The effects of generation, layer line, housing system, *post mortem* keel bone deformity and fracture and their interactions on the results of the bone mineral density (BMD) of the keel bone.

Effect	BMD (g/cm^2^)		Effect	BMD (g/cm^2^)	
	*F* Value	*p* -Value		*F* Value	*p* -Value
**Generation (G)**	39.13	<0.001	LL × HS	3.87	0.009
Layer line (LL)	14.62	<0.001	LL × KBD	0.286	0.944
Housing system (HS)	44.73	<0.001	LL × KBF	2.15	0.092
Keel bone deformity (KBD)	7.57	<0.001	HS × KBD	4.89	0.008
Keel bone fracture (KBF)	0.005	0.942	HS × KBF	1.72	0.189
G ×LL	2.36	0.074	KBD × KBF	1.56	0.210
G × HS	0.895	0.345	a	12.31	<0.001
G × KBD	1.44	0.238	b	0.284	0.889
G × KBF	1.15	0.283	c	0.422	0.793

Parameters of the Gompertz growth curve: a = adult body weight (g) of the hen at 69 weeks of age (asymptotic limit), b = slope of the curve, c = point of inflection (week).

**Table 5 animals-11-01546-t005:** Least Squares Means (LSM) their standard errors (SE) and significant differences between the effect levels for the bone mineral density (BMD) of the keel bone.

Effect	BMD (g/cm^2^)	Effect		BMD (g/cm^2^)
	LSM ± SE			LSM ± SE
**Generation**		**Layer line × Housing System**		
1st Generation	0.180 ^A^ ± 0.005	BLA	Floor housing	0.196 ^AB^ ± 0.009
2nd Generation	1.154 ^B^ ± 0.005		Cages	0.163 ^CD^ ± 0.005
**Layer Line**		L68	Floor housing	0.209 ^A^ ± 0.011
BLA	0.179 ^A^ ± 0.006		Cages	0.159 ^BCD^ ± 0.008
L68	0.184 ^A^ ± 0.009	WLA	Floor housing	0.159 ^CD^ ± 0.007
WLA	0.140 ^B^ ± 0.005		Cages	0.121 ^E^ ± 0.004
R11	0.165 ^A^ ± 0.007	R11	Floor housing	0.184 ^ABC^ ± 0.009
**Housing System**			Cages	0.147 ^D^ ± 0.006
Floor housing	0.187 ^A^ ± 0.007	**Housing System × KBD**		
Cages	0.147 ^B^ ± 0.003	Floor housing	Score 2	0.187 ^AB^ ± 0.007
**Keel Bone Deformities (KBD)**			Score 3	0.196 ^A^ ± 0.003
Score 2	0.167 ^B^ ± 0.004		Score 4	0.195 ^A^ ± 0.002
Score 3	0.177 ^A^ ± 0.003	Cages	Score 2	0.147 ^D^ ± 0.003
Score 4	0.183 ^A^ ± 0.003		Score 3	0.159 ^C^ ± 0.003
**Keel Bone Fractures (KBF)**			Score 4	0.171 ^B^ ± 0.003
Score 0	0.167 ^A^ ± 0.004			
Score 1	0.167 ^A^ ± 0.003			

^A–E^ Means within a column and effect with no common superscript differ significantly at *p* < 0.05.

**Table 6 animals-11-01546-t006:** The effects of the main factors, their interactions, and the covariates on the results of the bone mineral density (BMD) of the keel bone when hens were kept in cages.

Effect	BMD (g/cm^2^)		Effect	BMD (g/cm^2^)	
	*F* Value	*p* -Value		*F* Value	*p* -Value
Generation (G)	27.79	<0.001	LL × KBF	0.628	0.597
Layer line (LL)	2.17	0.091	KBD × KBF	0.441	0.644
Keel bone deformity (KBD)	7.92	<0.001	First egg	0.863	0.486
Keel bone fracture (KBF)	3.52	0.061	LP1	2.871	0.023
G × LL	0.702	0.552	LP2	6.550	<0.001
G × KBD	1.68	0.188	a	4.58	0.001
G × KBF	2.49	0.115	b	0.786	0.535
LL × KBD	1.4425	0.197	c	1.938	0.103

LP1 = laying performance from 18 to 29 weeks of age. LP2 = laying performance from 30 to 69 weeks of age. Parameters of the Gompertz growth curve: a = adult body weight (g) of the hen at 69 weeks of age (asymptotic limit), b = slope of the curve, c = point of inflection (weeks).

## Data Availability

The data presented in this study are available on reasonable request from the corresponding author.

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
