# Peer review of "Keel Bone Damage in Laying Hens—Its Relation to Bone Mineral Density, Body Growth Rate and Laying Performance"

_animals, 2021, doi:10.3390/ani11061546_

Round 1

Reviewer 1 Report

I would like to congratulate the authors for this in-depth review of animal welfare issues related to laying hens. Very interesting information that will definitely have application for future developments and housing considerations. 

Few comments/suggestions in the attached file. 

Author Response

Response to Reviewer 1

Dear Reviewer,

thank you for your very constructive feedback that helped us to improve our manuscript. In the following, you will find our responses to your comments point by point. We hope that the revision is satisfactory for you.

Yours sincerely, Christin Habig

1. Recommend to use euthanized as better word.

According to your recommendation we replaced “sacrifized” by “euthanized” (L16 and L154).

2. The abstract and summary are pretty much the same. Which I think allow for improvement. The abstract should contained more details on experimental design, treatments, stat analysis, results, p values, etc. I suggest authors to reword to better represent an Abstract given the authors guidelines which allow for a summary and abstract in independent sections.

We added some LSMs and p-values to the abstract, but due to the limited number of words (200) it is not possible to provide further details, without deletion of, in our opinion, important and necessary information.

3. No statement of Procedures being approved by an ethics committee for animal use in research.

The ‘Institutional Review Board Statement’ at the end of the manuscript (L548 to L550)contains this information.

4. I am going to be picky with this comment so this is to take with a grain of salt. I recommend to name pullet to the sexually immature laying hen. In this case is appropriate before moving to the laying house.

We replaced “hen” by “pullet” according to your comment in L101 and L106.

5. Please provide some diet / nutrient details of the feed program used throughout the trial.

As the composition of feeding stuffs have been already published by Jansen et al. (2020), we refer to their publication and add the following sentence to the material and methods section, L104 to L106:

“For further information on lighting regime, climatic conditions and the composition of feeding stuffs see Tables S1 and S2 in Jansen et al. [16].”

and L126 to L127

“Composition of layer diets is given in Table S3 in Jansen et al. [16].”

6. Any correction equation used in the data while analyzing the BMD??? This could be useful to avoid high variation in the numbers.

No correction equation was used while analyzing BMD.

Reviewer 2 Report

It is generally well written work concerning the actual problem of poultry production. I find the article interesting for the audience but some minor concerns need to be addressed before I can consider this manuscript acceptable for publication.

L142 – Are there any reference for this scoring system ?

L145-156 Not clear. How much time it took to scarify all hens ( or what was the number of hens selected from each pen everyday)  ?

L150 How KB were place on scanner? In which place KB were scanned ? It is a averaged result for a whole bone?

L170 Without explanation the interoperation of two other parameters of Gompertz function (b and c) the idea of application of this function is unclear. All factors included into models have should have interpretation. It must be explained to readers (growth rate constant and time of maximum growth rate). See L 350: without explanation, the true meaning of this sentence is unclear. Also in abstract (L36) you are writing about growth rate, which was not defined throughout the whole manuscript.

L229-232 As you explain in L386, you performed univariate regression analysis only for covariates found to significantly influence keels’ BMD. I think that is should be written here, in methods section, otherwise it is not clear why these three factors were selected

In Fig1  and Tables 1-3 footnotes please give the information of the n number in each group - figures and tables should be self-explanatory

Table 1-3 : statistical significance was calculated using chi-squared test?

L336 and 502: please do not write “denser” – for physical bodies density means mass/volume ratio, which was not determined. Please use “higher values of BMD”

Table 4 footnote: if you give the unit of a parameter (g), please also give units of b and c parameters (/wk and wk, respectively, if I’m not mistaken)

L383 Please consider showing these data not using normalized scientific notation but in notation with the same exponent integer (10^(-4)) - it will better show the differences between the values.

Figure 2 : x-axis labeling for A, C, E is misleading, regression coefficients are not measured in units of egg number of body weight. Please also correct “beta” to “beta1”. Finally, showed “standard errors of regression” are really S (standard error of regression) values,  which are measured in the units of the dependent variable (BMD)? Why didn’t you show CI of regression coefficient?

L493 – “white-egg layers” and “brown-egg layers”

L501 was the significance of this relationship measured? what was the R2 value of regression model? Was correlation coefficient (r) value significant ? What were p-values?

Reviewer 3 Report

This paper presents detailed measurements of bone mineral density and keel bone damage in four layer lines with different production level. Keel bone damage (both deviations and fractures) is compared between cage and floor housing. This study contributes to the understanding of one of the most prevalent welfare problems in laying hens and will be interesting to a wide audience.

The statistical models are well explained and the results are presented in tables and figures in a clear and understandable way. However, the description of housing is insufficient. Particularly, how big were the pens? Was there litter? What was the management? When you housed all 4 lines in one room what was you light regime? Was the light regime optimized for one of the lines? In particular, on lines 101 and 121: How much perch space was available per bird?

Why did you confound housing (cages, large pens, small pens) with material of perch (cages had a plastic perch and pens a metal perch) and height of perch (small pens had a low perch, large pens had a high perch)? Please account for the confounding of housing types and material of perch in the interpretation of the results.

How did you allocate the pullets to the pens? Were the lines mixed within pens?

Line 144: Was the palpator blind to the treatment (cage, large pen, small pen, line)? This is mentioned at the end of the paragraph for the dissected bones. Does it also apply to the palpations of live birds? What was the repeatability of palpations of live birds and the scoring of the bones? In that context, what was the correlation between the last palpations and the post-mortem scores?

Paragraph 2.4.3. Please specify which birds (caged, birds in pens) were analyzed with each model. Were caged birds only analyzed with model 3?

Did you pool non-significant interactions? If yes, what was your cut-off P-level for pooling?

Why did you not include the size of the pens into your models? Why did you choose different sized pens in the first place?

Line 272: Why were the generations analyzed separately and not generation added as a (random) factor? Wasn't generation something like a replicate?

Table 4: I am very surprised that KBF had no effect on BMD. Normally, fractures induce the creation of callus material which would increase BMD. What is your explanation for that? Did you detect callus material in the dissected bones? Or was there no healing of the fractures?

Line 475: … had not been equipped …

What is your explanation for the differences between the 2 generations?
